# AN IMAGE REPRESENTATION BASED CONVOLUTIONAL NETWORK FOR DNA CLASSIFICATION

Bojian Yin[1], Marleen Balvert[1,2], Davide Zambrano[1], Alexander Schönhuth[1,2,†], and Sander M. Bohte[1,†]

[1]*Centrum Wiskunde & Informatica (CWI), Amsterdam, The Netherlands*
[2]*Department of Biology, University of Utrecht, The Netherlands*
[†]*Joint last authorship*

## ABSTRACT

The folding structure of the DNA molecule combined with helper molecules, also referred to as the chromatin, is highly relevant for the functional properties of DNA. The chromatin structure is largely determined by the underlying primary DNA sequence, though the interaction is not yet fully understood. In this paper we develop a convolutional neural network that takes an image-representation of primary DNA sequence as its input, and predicts key determinants of chromatin structure. The method is developed such that it is capable of detecting interactions between distal elements in the DNA sequence, which are known to be highly relevant. Our experiments show that the method outperforms several existing methods both in terms of prediction accuracy and training time.

## 1 INTRODUCTION

DNA is perceived as a sequence over the letters $\{A,C,G,T\}$, the alphabet of nucleotides. This sequence constitutes the code that acts as a blueprint for all processes taking place in a cell. But beyond merely reflecting primary sequence, DNA is a molecule, which implies that DNA assumes spatial structure and shape. The spatial organization of DNA is achieved by integrating ("recruiting") other molecules, the histone proteins, that help to assume the correct spatial configuration. The combination of DNA and helper molecules is called chromatin; the spatial configuration of the chromatin, finally, defines the functional properties of local areas of the DNA (de Graaf & van Steensel, 2013).

Chromatin can assume several function-defining *epigenetic states*, where states vary along the genome (Ernst et al., 2011). The key determinant for spatial configuration is the underlying primary DNA sequence: sequential patterns are responsible for recruiting histone proteins and their chemical modifications, which in turn give rise to or even define the chromatin states. The exact configuration of the chromatin and its interplay with the underlying raw DNA sequence are under active research. Despite many enlightening recent findings (e.g. Brueckner et al., 2016; The EN-CODE Project Consortium, 2012; Ernst & Kellis, 2013), comprehensive understanding has not yet been reached. Methods that predict chromatin related states from primary DNA sequence are thus of utmost interest. In machine learning, many prediction methods are available, of which deep neural networks have recently been shown to be promising in many applications (LeCun et al., 2015). Also in biology deep neural networks have been shown to be valuable (see Angermueller et al. (2016) for a review).

Although DNA is primarily viewed as a sequence, treating genome sequence data as just a sequence neglects its inherent and biologically relevant spatial configuration and the resulting interaction between distal sequence elements. We hypothesize that a deep neural network designed to account for long-term interactions can improve performance. Additionally, the molecular spatial configuration of DNA suggests the relevance of a higher-dimensional spatial representation of DNA. However, due to the lack of comprehensive understanding with respect to the structure of the chromatin, sensible suggestions for such higher-dimensional representations of DNA do not exist.

One way to enable a neural net to identify long-term interactions is the use of fully connected layers. However, when the number of input nodes to the fully connected layer is large, this comes with a large number of parameters. We therefore use three other techniques to detect long-term interactions. First, most convolutional neural networks (CNNs) use small convolution filters. Using larger filters already at an early stage in the network allows for early detection of long-term interactions without the need of fully connected layers with a large input. Second, a deep network similar to the ResNet (He et al., 2015) or Inception (Szegedy et al., 2015) network design prevents features found in early layers from vanishing. Also, they reduce the size of the layers such that the final fully connected layers have a smaller input and don't require a huge number of parameters. Third, we propose a novel kind of DNA representation by mapping DNA sequences to higher-dimensional images using space-filling curves. Space-filling curves map a 1-dimensional line to a 2-dimensional space by mapping each element of the sequence to a pixel in the 2D image. By doing so, proximal elements of the sequence will stay in close proximity to one another, while the distance between distal elements is reduced.

The space-filling curve that will be used in this work is the Hilbert curve which has several advantages. **(i)**: [Continuity] Hilbert curves optimally ensure that the pixels representing two sequence elements that are close within the sequence are also close within the image (Bader, 2016; Aftosmis et al., 2004). **(ii)**: [Clustering property] Cutting out rectangular subsets of pixels (which is what convolutional filters do) yields a minimum amount of disconnected subsequences (Moon et al., 2001). **(iii)**: If a rectangular subimage cuts out two subsequences that are disconnected in the original sequence, chances are maximal that the two different subsequences are relatively far apart (see our analysis in Appendix A).

The combination of these points arguably renders Hilbert curves an interesting choice for representing DNA sequence as two-dimensional images. **(i)** is a basic requirement for mapping short-term sequential relationships, which are ubiquitous in DNA (such as codons, motifs or intron-exon structure). **(ii)** relates to the structure of the chromatin, which - without all details being fully understood - is tightly packaged and organized in general. Results from Elgin (2012) indicate that when arranging DNA sequence based on Hilbert curves, contiguous areas belonging to identical chromatin states cover rectangular areas. In particular, the combination of **(i)** and **(ii)** motivate the application of convolutional layers on Hilbert curves derived from DNA sequence: rectangular subspaces, in other words, submatrices encoding the convolution operations, contain a minimum amount of disconnected pieces of DNA. **(iii)** finally is beneficial insofar as long-term interactions affecting DNA can also be mapped. This in particular applies to so-called enhancers and silencers, which exert positive (enhancer) or negative (silencer) effects on the activity of regions harboring genes, even though they may be far apart from those regions in terms of sequential distance.

## 1.1 RELATED WORK

Since Watson and Crick first discovered the double-helix model of DNA structure in 1953 (Watson & Crick, 1953), researchers have attempted to interpret biological characteristics from DNA. DNA sequence classification is the task of determining whether a sequence $S$ belongs to an existing class $C$, and this is one of the fundamental tasks in bio-informatics research for biometric data analysis (Z. Xing, 2010). Many methods have been used, ranging from statistical learning (Vapnik, 1998) to machine learning methods (Michalski et al., 2013). Deep neural networks (LeCun et al., 2015) form the most recent class of methods used for DNA sequence classification (R.R. Bhat, 2016; Salimans et al., 2016; Zhou & Troyanskaya, 2015; Angermueller et al., 2016).

Both Pahm et al. (2005) and Higashihara et al. (2008) use support vector machines (SVM) to predict chromatin state from DNA sequence features. While Pahm et al. (2005) use the entire set of features as input to the SVM, Higashihara et al. (2008) use random forests to pre-select a subset of features that are expected to be highly relevant for prediction of chromatin state to use as input to the SVM. Only Nguyen et al. (2016) use a CCN as we do. There are two major differences between their approach and ours. First and foremost, the model architecture is different: the network in Nguyen et al. (2016) consists of two convolution layers followed by pooling layers, a fully connected layer and a sigmoid layer, while our model architecture is deeper, uses residual connections to reuse the learned features, has larger convolution filters and has small layers preceding the fully connected layers (see Methods). Second, while we use a space-filling curve to transform the sequence data into an image-like tensor, Nguyen et al. (2016) keep the sequential form of the input data.

Apart from Elgin (2012), the only example we are aware of where Hilbert curves were used to map DNA sequence into two-dimensional space is from Anders (2009), who demonstrated the power of Hilbert curves for visualizing DNA. Beyond our theoretical considerations, these last two studies suggest there are practical benefits of mapping DNA using Hilbert curves.

## 1.2 CONTRIBUTION

Our contributions are twofold. First, we predict chromatin state using a CNN that, in terms of architecture, resembles conventional CNNs for image classification and is designed for detecting distal relations. Second, we propose a method to transform DNA sequence patches into two-dimensional image-like arrays to enhance the strengths of CNNs using space-filling curves, in particular the Hilbert curve. Our experiments demonstrate the benefits of our approach: the developed CNN decisively outperforms all existing approaches for predicting the chromatin state in terms of prediction performance measures as well as runtime, an improvement which is further enhanced by the convolution of DNA sequence to a 2D image. In summary, we present a novel, powerful way to harness the power of CNNs in image classification for predicting biologically relevant features from primary DNA sequence.

## 2 METHODS

### 2.1 DNA SEQUENCE REPRESENTATION

We transform DNA sequences into images through three steps. First, we represent a sequence as a list of $k$-mers. Next, we transform each $k$-mer into a one-hot vector, which results in the sequence being represented as a list of one-hot vectors. Finally, we create an image-like tensor by assigning each element of the list of $k$-mers to a pixel in the image using Hilbert curves. Each of the steps is explained in further detail below.

From a molecular biology point of view, the nucleotides that constitute a DNA sequence do not mean much individually. Instead, nucleotide motifs play a key role in protein synthesis. In bioinformatics it is common to consider a sequence's $k$-mers, defined as the $k$-letter words from the alphabet {A,C,G,T} that together make up the sequence. In computer science the term $q$-gram is more frequently used, and is often applied in text mining (Tomovic et al., 2006). As an example, the sequence TGACGAC can be transformed into the list of 3-mers {TGA, GAC, ACG, CGA, GAC} (note that these are overlapping). The first step in our approach is thus to transform the DNA sequence into a list of $k$-mers. Previous work has shown that 3-mers and 4-mers are useful for predicting epigenetic state (Pahm et al., 2005; Higashihara et al., 2008). Through preliminary experiments, we found that $k = 4$ yields the best performance: lower values for $k$ result in reduced accuracy, while higher values yield a high risk of overfitting. Only for the Splice dataset (see experiments) we used $k = 1$ to prevent overfitting, as this is a small dataset.

In natural language processing, it is common to use word embeddings as GloVe or word2vec or one-hot vectors (Goldberg, 2017). The latter approach is most suitable for our method. Each element of such a vector corresponds to a word, and a vector of length $N$ can thus be used to represent $N$ different words. A one-hot vector has a one in the position that corresponds to the word the position is representing, and a zero in all other positions. In order to represent all $k$-mers in a DNA sequence, we need a vector of length $4^k$, as this is the number of words of length $k$ that can be constructed from the alphabet {A,C,G,T}. For example, if we wish to represent all 1-mers, we can do so using a one-hot vector of length 4, where A corresponds to [1 0 0 0], C to [0 1 0 0], G to [0 0 1 0] and T to [0 0 0 1]. In our case, the DNA sequence is represented as a list of 4-mers, which can be converted to a list of one-hot vectors each of length $4^4 = 256$.

Our next step is to transform the list of one-hot vectors into an image. For this purpose, we aim to assign each one-hot vector to a pixel. This gives us a 3-dimensional tensor, which is similar in shape to the tensor that serves as an input to image classification networks: the color of a pixel in an RGB-colored image is represented by a vector of length 3, while in our approach each pixel is represented by a one-hot vector of length 256.

What remains now is to assign each of the one-hot vectors in the list to a pixel in the image. For this purpose, we can make use of space-filling curves, as they can map 1-dimensional sequences

to a 2-dimensional surface preserving continuity of the sequence (Bader, 2016; Aftosmis et al., 2004). Various types of space-filling curves are available. We have compared the performance of several such curves, and concluded that Hilbert curves yield the best performance (Appendix A). This corresponds with our intuition: the Hilbert curve has several properties that are advantageous in the case of DNA sequences, as discussed in the introduction section.

The Hilbert curve is a well-known space-filling curve that is constructed in a recursive manner: in the first iteration, the curve is divided into four parts, which are mapped to the four quadrants of a square (Fig. 1a). In the next iteration, each quadrant is divided into four sub-quadrants, which, in a similar way, each hold 1/16$^{\text{th}}$ of the curve (Fig. 1b). The quadrants of these sub-quadrants each hold 1/64$^{\text{th}}$ of the curve, etc (Figs. 1c and d).

By construction, the Hilbert curve yields a square image of size $2^n \times 2^n$, where $n$ is the order of the curve (see Fig. 1). However, a DNA sequence does not necessarily have $2^n * 2^n$ $k$-mers. In order to fit all $k$-mers into the image, we need to choose $n$ such that $2^n * 2^n$ is at least the number of $k$-mers in the sequence, and since we do not wish to make the image too large, we pick the smallest such $n$. In many cases, a large fraction of the pixels then remains unused, as there are fewer $k$-mers than pixels in the image. By construction, the used pixels are located in upper half of the image. Cropping the picture by removing the unused part of the image yields rectangular images, and increases the fraction of the image that is used (Figure 1e).

In most of our experiments we used sequences with a length of 500 base pairs, which we convert to a sequence of 500 - 4 + 1 = 497 4-mers. We thus need a Hilbert curve of order 5, resulting in an image of dimensions $2^5 \times 2^5 \times 256 = 32 \times 32 \times 256$ (recall that each pixel is assigned a one-hot vector of length 256). Almost half of the resulting 1024 pixels are filled, leaving the other half of the image empty which requires memory. We therefore remove the empty half of the image and end up with an image of size $16 \times 32 \times 256$.

The data now has the appropriate form to input in our model.

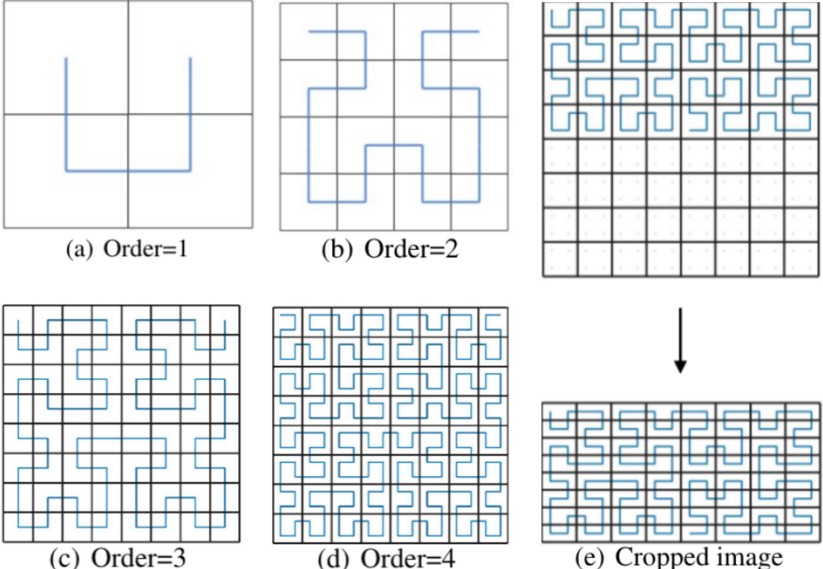

(a) Order=1    (b) Order=2

(c) Order=3    (d) Order=4    (e) Cropped image

**Figure 1:** Hilbert Curve and cropped image

## 2.2 NETWORK ARCHITECTURE

Modern CNNs or other image classification systems mainly focus on gray-scale images and standard RGB images, resulting in channels of length 1 or 3, respectively, for each pixel. In our approach, each pixel in the generated image is assigned a one-hot vector representing a $k$-mer. For increasing $k$, the length of the vector and thus the image dimension increases. Here, we use $k = 4$ resulting in

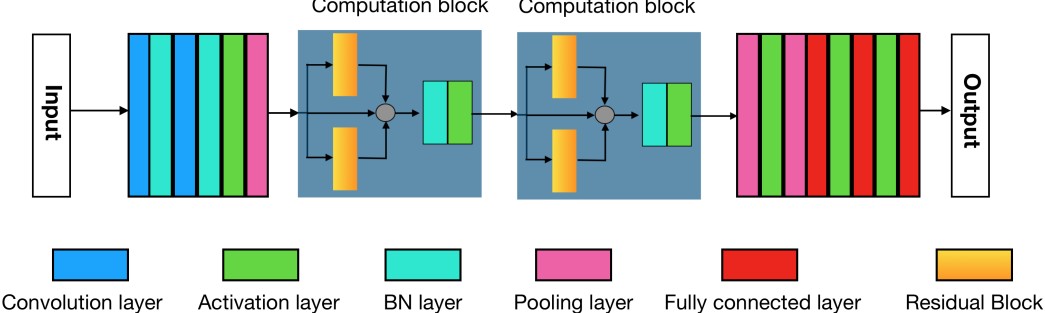

**Figure 2:** A simplified version of HCNN with two computational blocks. BN is short for "Batch Normalization".

256 channels, which implies that each channel contains very sparse information. Due to the curse of dimensionality standard network architectures applied to such images are prone to severe overfitting.

Here, we design a specific CNN for the kind of high dimensional image that is generated from a DNA sequence. The architecture is inspired by ResNet (He et al., 2015) and Inception (Szegedy et al., 2015). The network has $L$ layers and each layer implements a non-linear function $\mathcal{F}_l(x_l)$ where $l$ is the index of the hidden layer with output $x_{l+1}$. The function $\mathcal{F}_l(x_l)$ consists of various layers such as convolution (denoted by $c$), batch normalization ($bn$), pooling ($p$) and non-linear activation function ($af$).

The first part of the network has the objective to reduce the sparseness of the input image (Figure 2), and consists of the consecutive layers $[c, bn, c, bn, af, p]$. The main body of the network enhances the ability of the CNN to extract the relevant features from DNA space-filling curves. For this purpose, we designed a specific Computational Block inspired by the ResNet residual blocks (He et al., 2015). The last part of the network consists of 3 fully-connected layers, and softmax is used to obtain the output classification label. The complete model is presented in Table 1, and code is available on Github (https://github.com/Bojian/Hilbert-CNN/tree/master). A simplified version of our network with two Computational Blocks is illustrated in Figure 2.

**Computation Block.** In the Computation Block first the outputs of two Residual Blocks and one identity mapping are summed, followed by a $bn$ and an $af$ layer (Figure 2). In total, the computational block has 4 convolutional layers, two in each Residual Block (see Figure 3). The Residual Block first computes the composite function of five consecutive layers, namely $[c, bn, af, c, bn]$, followed by the concatenation of the output with the input tensor. The residual block concludes with an $af$.

The Residual Block can be viewed as a new kind of non-linear layer denoted by $Residual_l[k_j, k_{j+1}, d_{link}, d_{out}]$, where $k_j$ and $k_{j+1}$ are the respective filter sizes of the two convolutional layers. $d_{link}$ and $d_{out}$ are the dimensions of the outputs of the first convolutional layer and the Residual Block, respectively, where $d_{link} < d_{out}$; this condition simplifies the network architecture and reduces the computational cost. The Computational Block can be denoted as $\mathcal{C}[k1, k2, k3, k4]$ with the two residual blocks defined as $Residual_1[k_1, k_2, d_{link}, d_{out}]$ and $Residual_2[k_3, k_4, d_{link}, d_{out}]$. Note that here we chose the same $d_{link}$ and $d_{out}$ for both Residual Blocks in a Computational Block.

**Implementation details.** Most convolutional layers use small squared filters of size 2, 3 and 4, except for the layers in the first part of the network, where large filters are applied to capture long range features. We use Exponential Linear Units (ELU, Clevert et al. (2015)) as our activation function $af$ to reduce the effect of gradient vanishing: preliminary experiments showed that ELU preformed significantly better than other activation functions (data not shown).

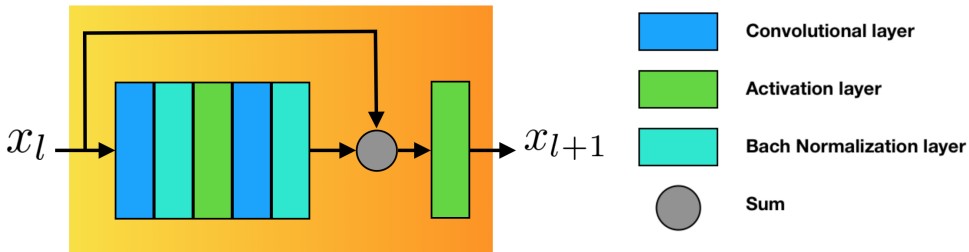

**Figure 3:** Residual Block

For the pooling layers $p$ we used Average pooling. Average pooling outperformed Max pooling in terms of prediction accuracy by more than $2\%$ in general, as it reduces the high variance of the sparse generated images. Cross entropy was used as the loss function.

**Table 1:** Network Architecture. The output size is given as height, width, channels.

| Layers | Description | Output size |
|---|---|---|
| Input | | $16, 32, 256$ |
| Conv1 | $7 \times 7$ conv, BN | $16, 32, 64$ |
| Conv2 | $5 \times 5$ conv, BN | $16, 32, 64$ |
| Activation | | |
| Average-Pool | $2 \times 2$ filter, stride 2 | $8, 16, 64$ |
| ComputationBlock | $[8,4,4,3], d_{link} = 4$ | $8, 16, 32$ |
| ComputationBlock | $[3,3,3,3], d_{link} = 4$ | $8, 16, 32$ |
| Average-Pool | $2 \times 2$ filter, stride 2 | $4, 8, 32$ |
| ComputationBlock | $[2,4,4,3], d_{link} = 4$ | $4, 8, 32$ |
| ComputationBlock | $[2,2,2,2], d_{link} = 4$ | $4, 8, 32$ |
| ComputationBlock | $[3,2,2,3], d_{link} = 4$ | $4, 8, 32$ |
| Average-Pooling | $2 \times 2$ filter, stride 2 | $2, 4, 32$ |
| BN,Activation | | |
| Average-Pool | $2 \times 2$ filter, stride 2 | $1, 2, 32$ |
| Classification layer1 | 64D FC layer | |
| | activation | |
| | dropout with 0.5 | |
| Classification layer2 | 100D FC layer | |
| | activation | |
| | dropout with 0.5 | |
| Classification layer3 | 50D FC layer, softmax | |

**Table 2:** DNA Datasets

| Name | ♯Samples | Description |
|---|---|---|
| H3 | 14965 | H3 occupancy |
| H4 | 14601 | H4 occupancy |
| H3K9ac | 27782 | H3K9 acetylation relative to H3 |
| H3K14ac | 33048 | H3K14 acetylation relative to H3 |
| H4ac | 34095 | H4 acetylation relative to H3 |
| H3K4me1 | 31677 | H3K4 monomethylation relative to H3 |
| H3K4me2 | 30683 | H3K4 dimethylation relative to H3 |
| H3K4me3 | 36799 | H3K4 trimethylation relative to H3 |
| H3K36me3 | 34880 | H3K36 trimethylation relative to H3 |
| H3K79me3 | 28837 | H3K79 trimethylation relative to H3 |
| Splice | 3190 | Splice-junction Gene Sequences |

## 3   EXPERIMENTS

We test the performance of our approach using ten publicly available datasets from Pokholok et al. (2005). The datasets contain DNA subsequences with a length of 500 base pairs. Each sequence is labeled either as "positive" or "negative", indicating whether or not the subsequence contains regions that are wrapped around a histone protein. The ten datasets each consider a different type of histone protein, indicated by the name of the dataset. Details can be found in Table 2.

A randomly chosen $90\%$ of the dataset is used for training the network, $5\%$ is used for validation and early stopping, and the remaining ($5\%$) is used for evaluation. We train the network using the AdamOptimizer (Kingma & Ba, 2017)[1]. The learning rate is set to 0.003, the batch-size was set to

---

[1]The LSTM model was implemented in Keras (Chollet et al., 2015), all other models were implemented in Tensorflow (Abadi et al., 2015).

300 samples and the maximum number of epochs is 10. After each epoch the level of generalization is measured as the accuracy obtained on the validation set. We use early stopping to prevent over-fitting. To ensure the model stops at the correct time, we combine the $GL_\alpha$ measurement (Prechelt, 1998) of generalization capability and the No-Improvement-In-N-Steps(Nii-N) method (Prechelt, 1998). For instance, Nii-2 means that the training process is terminated when generalization capability is not improved in two consecutive epochs.

We compare the performance of our approach, referred to as HCNN, to existing models. One of these is the support vector machine (SVM) model by Higashihara et al. (2008), for which results are available in their paper. Second, in tight communication with the authors, we reconstructed the Seq-CNN model presented in Nguyen et al. (2016) (the original software was no longer available), see Appendix C for detailed settings. Third, we constructed the commonly used LSTM, where the so-called 4-mer profile of the sequence is used as input. A 4-mer profile is a list containing the number of occurrences of all 256 4-mers of the alphabet {A,C,G,T} in a sequence. Preliminary tests showed that using all 256 4-mers resulted in overfitting, and including only the 100 most frequent 4-mers is sufficient. Details of the LSTM architecture can be found in Table 8 in Appendix C.

In order to assess the effect of using a 2D representation of the DNA sequence in isolation, we compare HCNN to a neural network using a sequential representation as input. We refer to this model as seq-HCNN. As in HCNN, the DNA sequence is converted into a list of kmer representing one-hot vectors, though the mapping of the sequence into a 2D image is omitted. The network architecture is a "flattened" version of the one used in HCNN: for example, a $7{\times}7$ convolution filter in HCNN is transformed to a $49{\times}1$ convolution filter in the 1D-sequence model. As a summary of model size, the Seq-CNN model contains 1.1M parameters, while both HCNN and seq-HCNN have 961K parameters, and the LSTM has 455K parameters.

In order to test whether our method is also applicable to DNA sequence classification tasks other than chromatin state prediction only, we performed additional tests on the splice-junction gene sequences dataset from Lichman (2013). Most of the DNA sequence is unused, and splice-junctions refer to positions in the genetic sequence where the transition from an unused subsequence (intron) to a used subsequence (exon) or vice versa takes place. The dataset consists of DNA subsequences of length 61, and each of the sequences is known to be an intron-to-exon splice-junction, an exon-to-intron splice junction or neither. As the dataset is relatively small, we used 1-mers instead of 4-mers. Note that the sequences are much shorter than for the other datasets, resulting in smaller images (dimensions $8 \times 8 \times 4$).

## 4 RESULTS

The results show that SVM and Seq-CNN were both outperformed by HCNN and seq-HCNN; LSTM shows poor performance. HCNN and seq-HCNN show similar performance in terms of prediction accuracy, though HCNN shows more consistent results over the ten folds indicating that using a 2D representation of the sequence improves robustness. Furthermore, HCNN yields better performance than seq-HCNN in terms of precision, recall, AP and AUC (Table 5). It thus enables to reliably vary the tradeoff between recall and false discoveries. HCNN outperforms all methods in terms of training time (Table 4).

The good performance of HCNN observed above may either be attributable to the conversion from DNA sequence to image, or to the use of the Hilbert curve. In order to address this question, we adapted our approach by replacing the Hilbert curve with other space-filling curves and compared their prediction accuracy. Besides the Hilbert curve, other space-filling curves exist (Moon et al., 2001) (see Appendix A ). In Figure 4, we compare the performance of our model with different mapping strategies in various datasets as displayed. We find that the images generated by the space-filling Hilbert curve provide the best accuracy on most datasets and the 1-d sequence performs worst.

## 5 DISCUSSION

In this paper we developed a CNN that outperforms the state-of-the-art for prediction of epigenetic states from primary DNA sequence. Indeed, our methods show improved prediction accuracy and training time compared to the currently available chromatin state prediction methods from Pahm

**Table 3:** Prediction accuracy obtained with an SVM-based method, Seq-CNN from Nguyen et al. (2016), LSTM, seq-HCNN and HCNN. The results for SVM are taken from Table 12 in Higashihara et al. (2008). In the splice dataset, Seq-CNN performed best when using 4-mers, while for HCNN and seq-HCNN 1-mers yielded the best performance.

| Dataset | SVM | LSTM | Seq-CNN | seq-HCNN | HCNN |
|---|---|---|---|---|---|
| H3 | 86.47% | 64.13% | 79.25% | 86.86 ± 1.563% | **87.34 ±0.263**% |
| H4 | 87.82% | 63.82% | 81.86% | 87.31 ± 0.952% | **87.33±0.264**% |
| H3K9ac | 75.08% | 63.07% | 68.76% | 78.47 ± 0.699% | **79.19±0.239**% |
| H3K14ac | 73.28% | 68.31% | 68.31% | **75.06 ±0.987**% | 74.79±0.226% |
| H4ac | 72.06% | 60.63% | 64.80% | 77.04 ± 1.256% | **77.06±0.233**% |
| H3K4me1 | 69.71% | 60.43% | 62.60% | **73.47 ±0.789**% | 73.21±0.221% |
| H3K4me2 | 68.97% | 61.45% | 62.38% | 73.91 ± 0.631% | **74.27±0.224**% |
| H3K4me3 | 68.57% | 58.03% | 62.33% | **74.54 ±0.865**% | 74.45±0.225% |
| H3K36me1 | 75.19% | 60.78% | 72.20% | **77.18 ±0.973**% | 77.03±0.232% |
| H3K79me1 | 80.58% | 63.84% | 75.07% | **81.66 ±1.264**% | 81.63±0.246% |
| Splice | 94.70% | 96.23% | 91.82% | 93.21 ± 1.645% | **94.11±0.284**% |

**Table 4:** Training times, presented as min:sec.

| Dataset | LSTM | seq-CNN | seq-HCNN | HCNN |
|---|---|---|---|---|
| H3 | 35:43 | 95:23 | 6:47 | 3:40 |
| H4 | 45:32 | 95:53 | 5:12 | 3:12 |
| H3K9ac | 76:06 | 173:18 | 17:24 | 7:40 |
| H3K14ac | 81:21 | 180:56 | 17:42 | 13:24 |
| H4ac | 93:32 | 181:33 | 24:48 | 17:32 |
| H3K4me1 | 93:44 | 192:20 | 18:30 | 10:38 |
| H3K4me2 | 94:22 | 188:13 | 18:23 | 14:38 |
| H3K4me3 | 96:03 | 162:32 | 20:40 | 11:33 |
| H3K36me3 | 93:48 | 161:12 | 21:52 | 16:37 |
| H3K79me3 | 64:28 | 158:34 | 14:25 | 10:13 |
| Splice | 6:42 | 35:12 | 3:42 | 1:30 |

et al. (2005), Higashihara et al. (2008) and Nguyen et al. (2016) as well as an LSTM model. Additionally, we showed that representing DNA-sequences with 2D images using Hilbert curves further improves precision and recall as well as training time as compared to a 1D-sequence representation.

We believe that the improved performance over the CNN developed by Nguyen et al. (2016) (Seq-CNN) is a result of three factors. First, our network uses larger convolutional filters, allowing the model to detect long-distance interactions. Second, despite HCNN being deeper, both HCNN and seq-HCNN have a smaller number of parameters, allowing for faster optimization. This is due to the size of the layer preceding the fully connected layer, which is large in the method proposed

**Table 5:** Recall, Precision, area under precision-recall curve (AP) and area under ROC curve (AUC) for seq-HCNN and HCNN. The reported values are the means over ten folds.

| Dataset | Recall | | Precision | | AP | | AUC | |
|---|---|---|---|---|---|---|---|---|
| | seq-HCNN | HCNN | seq-HCNN | HCNN | seq-HCNN | HCNN | seq-HCNN | HCNN |
| H3 | 85.67% | **87.33**% | 85.67% | **87.33**% | 90.33% | **93.33**% | 91.00% | **93.67**% |
| H4 | 87.00% | **87.33**% | 87.00% | **87.00**% | 92.67% | **94.67**% | 93.67% | **94.67**% |
| H3K9ac | 78.33% | **79.00**% | 78.67% | **79.00**% | 78.33% | **85.00**% | 79.67% | **85.33**% |
| H3K14ac | **74.00**% | 73.67% | 74.67% | **75.00**% | 73.67% | **79.67**% | 76.33% | **81.33**% |
| H4ac | 76.67% | **77.67**% | 77.33% | **78.33**% | 78.67% | **82.67**% | 80.33% | **83.33**% |
| H3K4me1 | 72.33% | **73.00**% | 72.67% | **73.67**% | 70.67% | **76.33**% | 71.67% | **78.33**% |
| H3K4me2 | 70.67% | **72.33**% | 73.00% | **74.00**% | 69.33% | **77.33**% | 70.00% | **78.67**% |
| H3K4me3 | 74.33% | **74.67**% | **75.00**% | 74.67% | 71.00% | **78.67**% | 72.00% | **80.00**% |
| H3K36me3 | 76.00% | **76.67**% | 77.00% | **77.67**% | 76.33% | **82.00**% | 79.33% | **83.00**% |
| H3K79me3 | 81.00% | **82.33**% | 81.00% | **82.67**% | 79.67% | **88.00**% | 81.00% | **88.67**% |
| Splice | 91.00% | **95.00**% | 90.67% | **94.33**% | 95.00% | **97.67**% | 97.33% | **98.67**% |

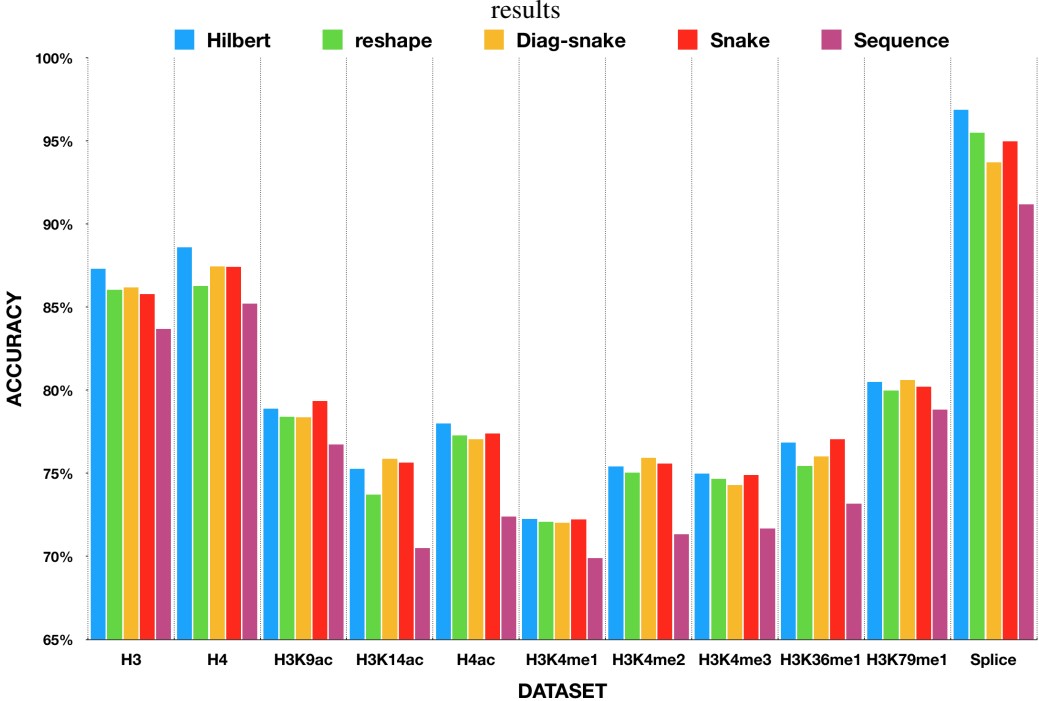

**Figure 4:** HCNN with different mapping strategies

by Nguyen et al. (2016) and thus yields a huge number of parameters in the fully connected layer. In HCNN on the other hand the number of nodes is strongly reduced before introducing a fully connected layer. Third, the use of a two-dimensional input further enhances the model's capabilities of incorporating long-term interactions.

We showed that seq-HCNN and HCNN are not only capable of predicting chromatin state, but can also predict the presence or absence of splice-junctions in DNA subsequences. This suggests that our approach could be useful for DNA sequence classification problems in general.

Hilbert curves have several properties that are desirable for DNA sequence classification. The intuitive motivation for the use of Hilbert curves is supported by good results when comparing Hilbert curves to other space-filling curves. Additionally, Hilbert curves have previously been shown to be useful for visualization of DNA sequences (Anders, 2009).

The main limitation of Hilbert curves is their fixed length, which implies that the generated image contains some empty spaces. These spaces consume computation resources; nevertheless, the 2D representation still yields reduced training times compared to the 1D-sequence representation, presumably due to the high degree of optimization for 2D inputs present in standard CNN frameworks.

Given that a substantial part of the improvements in performance rates are due to our novel architecture, we plan on investigating the details of how components of the architecture are intertwined with improvements in prediction performance in more detail. We also plan to further investigate why Hilbert curves yield the particular advantages in terms of robustness and false discovery control we have observed here.

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

## A    COMPARISON OF SPACE-FILLING CURVES WITH REGARD TO LONG-TERM INTERACTIONS

As noted before, long-term interactions are highly relevant in DNA sequences. In this section we consider these long-term interactions in four space-filling curves: the reshape curve, the snake curve, the Hilbert curve and the diag-snake curve. See Fig. 5 for an illustration.

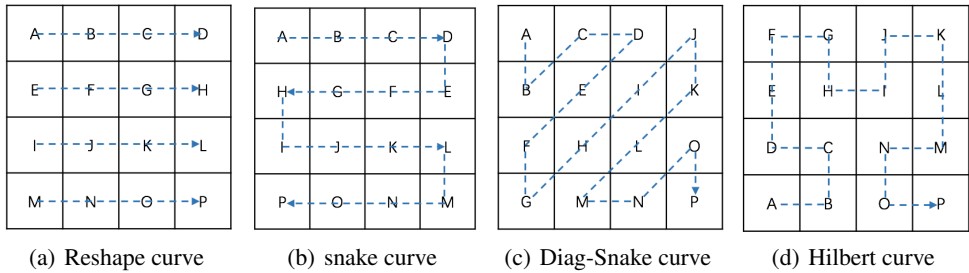

| (a) Reshape curve | (b) snake curve | (c) Diag-Snake curve | (d) Hilbert curve |

**Figure 5:** Space-filling curves

As can be seen in Fig. 5, mapping a sequence to an image reduces the distance between two elements that are far from one another in the sequence, while the distance between nearby elements does not increase. Each of the curves does have a different effect on the distance between far-away elements. In order to assess these differences, we use a measure that is based on the distance between two sequence elements as can be observed in the image. We denote this distance by $\mathcal{L}_C(x,y)$ where $x, y \in \mathcal{S}$, with $\mathcal{S}$ the sequence and $C$ the curve under consideration. Then for the sequence $\mathcal{S} = \{A,B,C,D,E, \cdots ,P\}$ we obtain

- $\mathcal{L}_{seq}(A,P) = 15$ for the sequence;
- $\mathcal{L}_{reshape}(A,P) = 3\sqrt{2}$ for the reshape curve;
- $\mathcal{L}_{snakesnake}(A,P) = 3$ for the snake curve;
- $\mathcal{L}_{diag-snake}(A,P) = 3\sqrt{2}$ for the diagonal snake curve.
- $\mathcal{L}_{hilbert}(A,P) = 3$ for the Hilbert curve;

We now introduce the following measure:

$$\Gamma(\mathbf{C}) = \frac{\text{mean}(\Delta(\mathbf{C}))}{\text{max}(\Delta(\mathbf{C}))},$$

where the $\Delta(\mathbf{C})$ is the set of the weighted distances between all pairs of the elements in the sequence. Here, $\Delta(C)$ is a set containing the distance between any two sequence elements, weighted by their distance in the sequence:

$$\Delta(\mathbf{C}) = \{\mathcal{L}_{seq}(x,y) \cdot \mathcal{L}_{\mathbf{C}}(x,y) \mid x, y \in \mathbf{C}, x \neq y\}.$$

Note that a low $\text{max}(\Delta(\mathbf{C}))$ relative to $\text{mean}(\Delta(\mathbf{C}))$ implies that long-term interactions are strongly accounted for, so a high $\Gamma(C)$ is desirable.

$\Gamma(C)$ is evaluated for the four space-filling curves as well as the sequence representation for sequences of varying lengths.The results show that the Hilbert curve yields the highest values for $\Gamma(C)$ 6 and thus performs best in terms of retaining long-distance interactions.

| Curve | Sequence Length : bits | | | | |
|---|---|---|---|---|---|
| | 16 | 64 | 256 | 1024 | 4096 |
| Reshape | 0.22 | 0.18 | 0.16 | 0.16 | 0.15 |
| Snake | 0.28 | 0.20 | 0.17 | 0.16 | 0.16 |
| Diag-snake | 0.22 | 0.17 | 0.16 | 0.15 | 0.15 |
| Hilbert | **0.30** | **0.24** | **0.22** | **0.21** | **0.21** |

**Table 6:** $\Gamma(C)$ for four space-filling curves, evaluated in sequences of varying lengths.

## B  FROM SEQUENCE TO IMAGE

Figure 6 shows the conversion from DNA sequence to image.

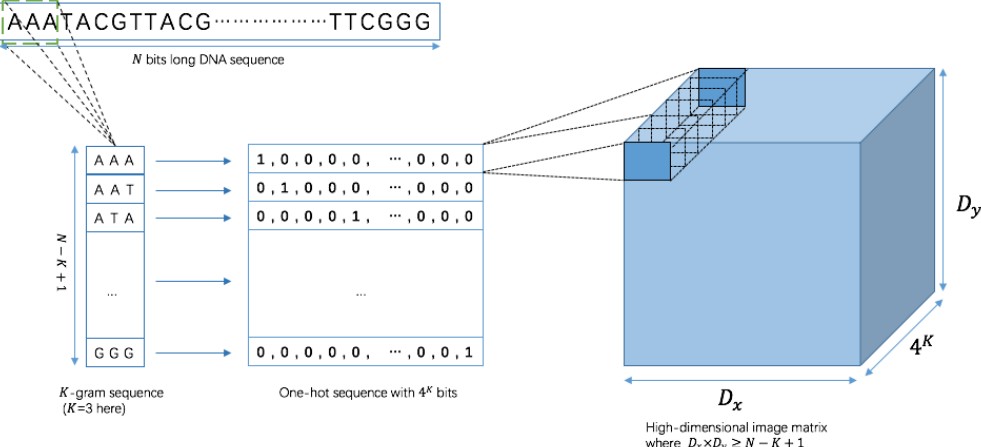

**Figure 6:** Sequence to Image

## C  DETAILS OF ALTERNATIVE NEURAL NETWORKS

**Table 7:** The model architecture of seq-CNN (Nguyen et al., 2016)

| Layer | filter size | stride | output dim |
|---|---|---|---|
| Convolution 1 | 7 | 2 | 60 |
| Activation | | | |
| Max pooling | 3 | 3 | 60 |
| Convolution 2 | 5 | 2 | 30 |
| Activation | | | |
| Max pooling | 3 | 2 | 30 |
| Dropout, 0.5 | | | |
| FC layer | | | 100 |
| Activation | | | |
| Dropout, 0.5 | | | |
| Classifier | | | 2 |

**Table 8:** Bir-Direction LSTM

| Layer | Description |
|---|---|
| Embedding | |
| Conv 1 | 1-by-3 convolution layer with 32 filter size activation function is RELU |
| Max pooling | $1 \times 2$ max pooling layer |
| Bir-LSTM 1 | 100 units |
| Bir-LSTM 2 | 128 units |
| Dropout | 0.3 dropout rate |
| Classifier | sigmoid |

| Architecture | # Parameters |
|---|---|
| seq-CNN | 1.1M |
| biLSTM | 455K |
| Hilbert-CNN | 961K |

**Table 9:** Table with the number of network parameters

## D  HYPERPARAMETER OPTIMIZATION

Accuracy is one of the most intuitive performance measurements in deep learning and machine learning. We therefore optimized the hyperparameters such as the network architecture and learning rate based on maximum accuracy. The hyperparameters are optimized through random search (Bergstra & Bengio, 2012) and using general principles from successful deep learning strategies and the following intuition. First, as our main goal was to capture long-term interactions, we chose a large kernel size for the first layer, for which various values were attempted and 7x7 gave the best performance. As is common practice in deep learning, we then opted for a smaller kernel size in the following layer. Second, in order to limit the number of parameters, we made use of residual blocks inspired by ResNet (He et al., 2015) and Inception (Szegedy et al., 2015). Finally, we applied batch normalization to prevent overfitting.

