# OpenReview forum: "An image representation based convolutional network for DNA classification"
_ICLR.cc/2018/Conference — Accept (Poster)_

### Official Review · AnonReviewer2 · 2017-11-22
**interesting method**

**Rating:** 7
**Confidence:** 3

**Review:**

There are functional elements attached on the DNA sequence, such as transcription factors and different kinds of histones as stated in this ms. A hidden assumption is that the binding sites of these functional elements over the genome share some common features. It is therefore biologically interesting to predict if a new DNA sequence could be a binding site. Naturally this is is classification problem where the input is the DNA sequence and the output is whether the give sequence is a binding site.

This ms makes a novel way to transform the DNA sequence into a 3-dimensional tensor which could be easily utilised by CNN for images. The DNA sequence is first made into a a list of 4-mers. Then then each 4-mer is coded as a 4^4=256 dimensional vector. The order of the 4-mers is then coded into a image using Hilbert curve which presumably has nice properties to keep spatial information.

I am not familiar with neural networks and do not comment on the methods but rather from the application point of view.

First to my best knowledge, it is still controversial if the binding sites of different histones carries special features. I mean it could be possible that the assumption I mentioned in the beginning may not hold for this special application, especially for human data. I feel this method is more suitable for transcription factor motif data. see https://www.nature.com/articles/nbt.3300

Second, the experiments data in 2005 is measured using microarray, which uses probes of 500bp long. But the whole binding site for a nucleosome (or histone complex) is 147bp, which is much shorter than the probe. Nowadays we have more accurate sequencing data for nucleosome (check https://www.ncbi.nlm.nih.gov/pubmed/26411474). I am not sure whether this result will generalised to some other similar dataset.

Third, the results only list the accuracy, it will be interesting to see the proportion of false negatives.

In general I feel the transformation is quite useful, it nicely reserves the spatial information, also can be seen from the improved results over all datasets. The result, in my opinion, is not sufficient to support the assumption that we could predict the DNA structures solely base on the sequence.

---

### Official Review · AnonReviewer1 · 2017-11-22
**Innovative but insufficiently evaluated**

**Rating:** 7
**Confidence:** 5

**Review:**

Dear editors,

the authors addressed all of my comments and clearly improved their manuscript over multiple iterations. I therefore increased my rating from ‘6: Marginally above acceptance threshold’ to ‘7: Good paper, accept’.
Please note that the authors made important edits to their manuscript after the ICLR deadline and could hence not upload their most current version, which you can from https://file.io/WIiEw9. If you decided to publish the manuscript, I hence highly suggest using this (https://file.io/WIiEw9) version.

Best,

--------------------------------------------------------------------------------------------------------------------------------------------------------------------------------------------

The authors present Hilbert-CNN, a convolutional neural network for DNA sequence classification. Unlike existing methods, their model does not use the raw one-dimensional (1D) DNA sequence as input, but two-dimensional (2D) images obtained by mapping sequences to images using spacing-filling Hilbert-Curves. They further present a model (Hilbert-CNN) that is explicitly designed for Hilbert-transformed DNA sequences. The authors show that their approach can increase classification accuracy and decrease training time when applied to predicting histone-modification marks and splice junctions.

Major comments
=============
1. The motivation of transforming sequences into images is unclear and claimed benefits are not sufficiently supported by experiments. The essence of deep neural networks is to learn a hierarchy of features from the raw data instead of engineering features manually. Using space filling methods such as Hilbert-curves to transform (DNA) sequences into images can be considered as unnecessary feature-engineering.

The authors claim that ‘CNNs have proven to be most powerful when operating on multi-dimensional input, such as in image classification’, which is wrong. Sequence-based convolutional and recurrent models have been successfully applied for modeling natural languages (translation, sentiment classification, …), acoustic signals (speech recognition, audio generation), or biological sequences (e.g. predicting various epigenetic marks from DNA as reviewed in Angermueller et al).

They further claim that their method can ‘better take the spatial features of DNA sequences into account’ and  can better model ‘long-term interactions’ between distant regions. This is not obvious since Hilbert-curves map adjacent sequence characters to pixels that are close to each other as described by the authors, but distant characters to distant pixels. Hence, 2D CNN must be deep enough for modeling interactions between distant image features, in the same way as a 1D CNN.

Transforming sequences to images has several drawbacks. 1) Since the resulting images have a small width and height but many channels, existing 2D CNNs such as ResNet or Inception can not be applied, which also required the authors to design a specific model (Hilbert-CNN). 2) Hilbert-CNN requires more memory due to empty image regions. 3) Due to the high number of channels, convolutional filters have more parameters. 4) The sequence-to-image transformation makes model-interpretability hard, which is in particular important in biology. For example, motifs of the first convolutional layers can not be interpreted as sequence motifs (as described in Angermueller et al) and it is unclear how to analyze the influence of sequence characters using attention or gradient-based methods.

The authors should more clearly motivate their model in the introduction, tone-down the benefit of sequence-to-image transformations, and discuss drawbacks of their model. This requires major changes of introduction and discussion.

2. The authors should more clearly describe which and how they optimized hyper-parameters. The authors should optimize the most important hyper-parameters of their model (learning rate, batch size, weight decay, max vs. average pooling, ELU vs. ReLU, …) and baseline models on a holdout validation set. The authors should also report the validation accuracy for different sequence lengths, k-mer sizes, and space filling functions. Can their model be applied to longer sequences (>= 1kbp) which had been shown to improve performance (e.g. 10.1101/gr.200535.115)? Does Figure 4 show the performance on the training, validation, or test set?

3. It is unclear if the performance gain is due the proposed sequence-to-image transformation, or due to the proposed network architecture (Hilbert-CNN). It is also unclear if Hilbert-CNNs are applicable to DNA sequence classification tasks beyond predicting chromatin states and splice junctions. To address these points, the authors should compare Hilbert-CNN to models of the same capacity (number of parameters) and optimize hyper-parameters (k-mer size, convolutional filter size, learning rate, …) in the same way as they did for Hilbert-CNN. The authors should report the number of parameters of all models (Hilbert-CNN, Seq-CNN, 1D-sequence-CNN (Table 5), and LSTM (Table 6), …) in an additional table. The authors should also compare Hilbert-CNN to the DanQ architecture on predicting epigenetic markers using the same dataset as reported in the DanQ publication (DOI: 10.1093/nar/gkw226). The authors should also compare Hilbert-CNNs to gapped-kmer SVM, a shallow model that had been successfully applied for genomic prediction tasks.

4. The authors should report the AUC and area under precision-recall curve (APR) in additional to accuracy (ACC) in Table 3.

5. It is unclear how training time was measured for baseline models (Seq-CNN, LSTM, …). The authors should use the same early stopping criterion as they used for training Hilber-CNNs. The authors should also report the training time of SVM and gkm-SVM (see comment 3) in Table 3.


Minor comments
=============
1. The authors should avoid uninformative adjectives and clutter throughout the manuscript, for example ‘DNA is often perceived’, ‘Chromatin can assume’,  ‘enlightening’, ‘very’, ‘we first have to realize’, ‘do not mean much individually’, ‘very much like the tensor’, ‘full swing’, ‘in tight communication’, ‘two methods available in the literature’.

The authors should point out in section two that k-mers can be overlapping.

2. Section 2.1: One-hot vectors is not the only way for embedding words. The authors should also mention Glove and word2vec. Similar approaches had been applied to protein sequences (DOI: 10.1371/journal.pone.0141287)

3. The authors should more clearly describe how Hilbert-curves map sequences to images and how images are cropped. What does ‘that is constructed in a recursive manner’ mean? Simply cropping the upper half of Figure 1c would lead to two disjoint sequences. What is the order of Figure 1e?

4. The authors should consistently use ‘channels’ instead of ‘full vector of length’ to denote the dimensionality of image pixels.

5. The authors should use ‘Batch norm’ instead of ‘BN’ in Figure 2 for clarification.

6. Hilber-CNN is similar to ResNet (DOI: 10.1371/journal.pone.0141287), which consists of multiple ‘residual blocks’, where each block is a sequence of ‘residual units’. A ‘computational block’ in Hilbert-CNN contains two parallel ‘residual blocks’ (Figure 3) instead of a sequence of ‘residual units’. The authors should use ‘residual block’ instead of ‘computational block’, and ‘residual units’ as in the original ResNet publication. The authors should also motivate why two residual units/blocks are applied parallely instead of sequentially.

7. Caption table 1: the authors should clarify if ‘Output size’ is ‘height, width, channels’, and explain the notation in ‘Description’ (or refer to the text.)

---

### Official Review · AnonReviewer3 · 2017-11-25
**The idea of converting k-mer representations to 2D images using Hilbert curves is novel in application.**

**Rating:** 7
**Confidence:** 5

**Review:**

The authors of this manuscript transformed the k-mer representation of DNA fragments to a 2D image representation using the space-filling Hilbert curves for the classification of chromatin occupancy. In generally, this paper is easy to read. The components of the proposed model mainly include Hilbert curve theory and CNN which are existing technologies. But the authors make their combination useful in applications. Some specific comments are:

1. In page 5, I could not understand the formula d_kink < d_out. d_link ;
2. There may exist some new histone modification data that were captured by the next-generation sequencing (e.g. ChIP-seq) and are more accurate;
3. It seems that the authors treat it as a two-class problem for each data set. It would be more useful in real applications if all the data sets are combined to form a multi-class problem.

---

### Public Comment · (anonymous) · 2017-12-03
**Validation of method**

Hi authors,

The usage of a Hilbert space filling curve for chromatin structure classification is not clearly motivated to me. If I understand the method correctly, the goal is to perform some deterministic mapping from the 1D structure to the 2D space filling curve in order to leverage the spatial structure of popular CNN methods. However, since the Hilbert space filling curve does not leverage any biologically motivated information to construct the 2D image, it is unclear how this would help. The Hilbert space filling curve has nice properties such as (continuity, clustering, etc), but the original 1D sequence also possesses all of these properties. It seems simply using a larger filter size on the 1D sequence should achieve similar results since the 1D sequence itself should be able to achieve the continuity, clustering, etc properties of the Hilbert space filling curve.

In addition, as the below reviewers have mentioned, the success of CNNs are not specific to 2D images, and there is not really a theoretical argument for converting a 1D structure to a 2D structure. Am I missing something here regarding the problem of chromatin structure classification or the representational power of Hilbert space filling curves?

Furthermore, the results section has not fully convinced me that the Hilbert-CNN is more powerful than applying a direct 1D CNN on the sequence. The method compared against in Nguyen et al (2016) only uses a 7 x 1 filter size in the first layer while the Hilbert-CNN uses a 7x7 filter size in the first layer. Perhaps a more convincing comparison would be comparing against a method that uses the same number of parameters (i.e. a 1D method that uses a filter size of 49 x 1 in the first layer) to demonstrate the utility of converting a 1D sequence to a 2D image.

I would be very interested if the Hilbert space filling curve was indeed a superior representation than its 1D counterpart.

---

> ### Author Response · Authors · 2017-12-12
> **Bigger 1D model improves while 2D Hilbert curves are faster and more consistent**
>
> Dear reviewer,
>
> Thank you very much for your comment. We feel that you have made an excellent point here regarding the filter-size difference between the 1D-sequence CNN and the Hilbert CNN.  Your suggestion has resulted in further insight into the particular advantages of the components of our approach, which, to our feeling, has led to non-negligible improvements of the manuscript.
>
> First, one comment on properties: while continuity and clustering properties are trivially preserved by keeping 1D sequence, the third property we are citing is not preserved by 1D sequence. Namely, different, separate subsequences going into the filter, are, on average, farthest apart within the sequence for Hilbert curves. So, beyond considering larger filter sizes, Hilbert curves can still have particular advantages because of mapping distal relationships (and are even optimal in that regard).
>
> To investigate the interplay of the novelties we have been suggesting further, we ran the 1D-sequence CNN now with an identical number of parameters in all layers (e.g. a 7x7 convolution in the first layer of Hilbert-CNN corresponds to a 49x1). In addition, following a suggestion from one of the other reviewers, we computed recall, precision, area under precision-recall curve (AP) and AUC, where precision-recall and ROC curves were computed by filtering the softmax output. The corresponding additional results can be found in the updated version of the document, in tables 3, 4 and 5.
>
> We find that :
> 1. Both the 1D sequence input and the 2D image input significantly outperform the current state of the art (which, as we feel, justifies that our paper is discussed at a venue such as the ICLR).
> 2. We further observe that the standard deviation of the networks across different training runs is much smaller for the 2D (e.g. 7x7) Hilbert curve representation than for the naive 1D (e.g. 49x1) one.
> We find that the 2D representation yields improvement over the 1D sequence representation in terms of recall and precision, and, in terms of AP and AUC, even drastic improvements (raising performance by more than 5% on average over the 49x1 filters).
>
> We conclude that:
> 1. In terms of basic performance measures such as accuracy, recall, precision, the architecture of our network (which in comparison to prior work avoids large layers to precede the fully connected layers) is the decisive factor.
> 2. Nevertheless, Hilbert curves enable
> a. More robust predictions, as indicated by the significantly lower variance in accuracy across different runs.
> b. Filtering of the output for optimizing in terms of precision-recall - tradeoffs (which the 1D output does not reliably allow to do).
>
> Overall, these findings suggest that there is particular advantage with respect to using 2D representations in the form of space filling (in particular Hilbert) curves, while there is also substantial power in the proposed network structure (which, as a novelty, more decidedly reduces numbers of hidden nodes in layers preceding the fully connected layers).
>
> We reworked the introduction and discussion to more properly reflect this finding and adapted the title accordingly.

---

### Public Comment · ~Akash_Singh1 · 2017-12-16
**On reproducing “Convolving DNA using two-dimensional Hilbert curve representations”[1]**

Hello authors,

We found this paper really interesting and therefore took this opportunity to fully understand your work by reproducing the results. Please refer to the link below to review the challenges that one can face while using your proposed paper for further innovations.
Link:

https://drive.google.com/file/d/1ged4-Yxisl8HKltGEvBMf936BZCJ4SQZ/view?usp=sharing

Thanks and regards
Akash Singh
akash.singh@mail.mcgill.ca
(on behalf of team)

---

> ### Author Response · Authors · 2017-12-22
> **Differences between the two approaches**
>
> Dear Akash and team,
>
> Thank you for your interest in our paper, we appreciate your effort in reproducing the results.
>
> We thoroughly read your report, and noted some differences between your implementation and ours which are likely to have caused the differences in results. First, the early stopping approach that is used differs: the use of GL0 stops the training process earlier than then the GL2 we used. Additionally, we used a combination of GL2 and No-improvement-in-N-steps. Second, the output is of a different form: instead of using 0/1 values for class prediction, we used one-hot vectors as the output. This generally improves prediction accuracy (see also https://stackoverflow.com/questions/17469835/why-does-one-hot-encoding-improve-machine-learning-performance). Third, we used self-defined droput and normalization which improves efficiency and accuracy. Finally, our k-mer representation is indeed different from the one you are using: the value of a k-mer is based on the occurrence of the subsequence in the dataset (higher occurence - lower number (Bojian: or higher?)). The latter however is probably not causing a significant difference, according to some tests we ran. We will add the missing information in the next version of our manuscript in as far as space allows.
>
> We hope this answers your questions. If you have any further questions, feel free to contact us.

---

### Decision · Program_Chairs · 2018-01-29
**ICLR 2018 Conference Acceptance Decision**

**Decision:**

Accept (Poster)

**Comment:**

This paper addresses an important application in genomics, i.e. the prediction of chromatin structure from nucleotide sequences.  The authors develop a novel method for converting the nucleotide sequences to a 2D structure that allows a CNN to detect interactions between distant parts of the sequence.  The reviewers found the paper innovative, interesting and convincing.  Two reviewers gave a 7 and there was one 6.  The 6, however, indicated during rather lengthy discussion that they were willing to raise their scores if their comments were addressed.  Hopefully the authors will address these comments in the camera ready version.  Overall a solid application paper with novel insights and technical innovation.